# Primary series COVID-19 vaccine effectiveness among health care workers in the country of Georgia, March–December 2021

Mark A. Katz[1]⊕*, Madelyn Yiseth Rojas Castro[2]⊕, Giorgi Chakhunashvili[3], Nazibrola Chitadze[3], Caleb L. Ward[4], C. Jason McKnight[5], Héloïse Lucaccioni[6], Iris Finci[1], Tamila Zardiashvili[7], Richard Pebody[1], Esther Kissling[2], Lia Sanodze[3]

1 World Health Organization Regional Office for Europe, Copenhagen, Denmark, 2 Epiconcept, Paris, France, 3 National Center for Disease Control and Public Health, Tbilisi, Georgia, 4 Public Health Institute, Tbilisi, Georgia, 5 South Caucasus Hub, World Health Organization Regional Office for Europe, Tbilisi, Georgia, 6 European Programme for Intervention Epidemiology Training, European Centre for Disease Prevention and Control, Stockholm, Sweden, 7 WHO Country Office in Georgia, Tbilisi, Georgia

⊕ These authors contributed equally to this work.
* katzm@who.int

**Data Availability Statement:** All relevant data are within the manuscript and its Supporting information files. Data beyond what is included in

## Abstract

### Background

Healthcare workers (HCWs) have suffered considerable morbidity and mortality during the COVID-19 pandemic. Few data on COVID-19 vaccine effectiveness (VE) are available from middle-income countries in the WHO European Region. We evaluated primary series COVID-19 VE against laboratory-confirmed COVID-19 among HCWs in Georgia.

### Methods

HCWs in six hospitals in Georgia were invited to enroll in a prospective cohort study conducted during March 19–December 5, 2021. Participants completed weekly symptom questionnaires. Symptomatic HCWs were tested by RT-PCR and/or rapid antigen test (RAT), and participants were routinely tested for SARS-CoV-2 by RT-PCR or RAT, regardless of symptoms. Serology was collected at enrolment, and quarterly thereafter, and tested by electrochemiluminescence immunoassay for SARS-CoV-2 antibodies. We defined primary series vaccination as two doses of COVID-19 vaccine received ≥14 days before symptom onset. We estimated VE as (1-hazard ratio)*100 using a Cox proportional hazards model with vaccination status as a time-varying covariate. Estimates were adjusted by potential confounders that changed the VE estimate by more than 5%, according to the change-in-estimate approach.

### Results

Overall, 1561/3849 (41%) eligible HCWs enrolled and were included in the analysis. The median age was 40 (IQR: 30–53), 1318 (84%) were female, and 1003 (64%) had laboratory evidence of prior SARS-Cov-2 infection. At enrolment, 1300 (83%) were unvaccinated; By study end, 1082 (62%) had completed a primary vaccine series (69% BNT162b2 (Pfizer-

the manuscript cannot be shared publicly because public deposition would breach compliance with the protocol, and the informed consent form, approved by the two research ethics boards that reviewed the protocol.

**Funding:** This study was supported by funding from the World Health Organization/European Regional Office, the US Centers for Disease Control and Prevention (Cooperative Agreement No.: NU2GGH002093), and the Public Health Institute. WHO/European Regional office staff were involved in study design, data collection and analysis, decision to publish, and preparation of the manuscript. The US CDC and the Public Health Institute had no role in in study design, data collection and analysis, decision to publish, or preparation of the manuscript. Three authors (GC, NC, and LS) received stipends from the funding institutions in order to carry out the study in Georgia.

**Competing interests:** None of the authors report any conflicts of interest.

BioNTech); 22% BBIBP-CorV (Sinopharm); 9% other). During the study period, 191(12%) participants had a new PCR- or RAT-confirmed symptomatic SARS-CoV-2 infection. VE against PCR- or RAT- confirmed symptomatic SARS-CoV-2 infection was 58% (95%CI: 41; 70) for all primary series vaccinations, 68% (95%CI: 51; 79) for BNT162b2, and 40% (95% CI: 1; 64) for BBIBP-CorV vaccines. Among previously infected HCWs, VE was 58% (95% CI: 11; 80). VE against medically attended COVID-19 was 52% (95%CI: 28; 68), and VE against hospitalization was 69% (95% CI: 36; 85). During the period of predominant Delta variant circulation (July-December 2021), VE against symptomatic COVID-19 was 52% (95%CI: 30; 66).

## Conclusions

Primary series vaccination with BNT162b2 and BBIBP-CorV was effective at preventing COVID-19 among HCWs, most of whom had previous infection, during a period of mainly Delta circulation. Our results support the utility of COVID-19 primary vaccine series, and the importance of increasing coverage, even among previously infected individuals.

## Introduction

Health care workers (HCWs) have suffered high rates of morbidity and mortality during the COVID-19 pandemic [1]. During the COVID pandemic and all pandemics, it is critical to protect the health of HCWs for a number of reasons; [2–4] first, in order to ensure the continuous functioning of the healthcare system, a critical essential service in the pandemic response, and one which is particularly challenging in resource-poor healthcare systems in low- and middle-income countries (LMICs). [5]; second, HCWs may have greater likelihood of getting infected with SARS-CoV-2 and other pathogens compared to the general population because of their close contact with infected patients; third, infected HCWs in healthcare settings risk transmitting viruses to vulnerable patients; finally, protecting HCWs is important because of the principle of reciprocity; through their vital front-line role in the pandemic response, HCWs put themselves at risk and also potentially put their households at higher risk for the sake of others [4].

In addition, HCWs play a vital role in the planning, coordination and execution of vaccination campaigns in high-income countries and LMICs. Population-wide COVID-19 vaccination campaigns have been shown to reduce pressure on the healthcare systems [6].

COVID-19 vaccine has been a critical intervention to reduce both morbidity and mortality during the pandemic. Since late 2020, when global vaccine distribution began, nearly 13.5 billion COVID-19 vaccine doses have been administered worldwide [7]. In the first year after their introduction, COVID-19 vaccines were estimated to have saved nearly 20 million lives globally [8], and over 440,000 lives among persons $\geq$ 60 years old in Europe [9].

Understanding COVID-19 vaccine effectiveness is critical to inform national and international vaccination guidelines. However, despite the extensive use of COVID-19 vaccines in LMICs, few VE studies have been conducted in LMICs. In the WHO European Region, where 17 of the 53 member countries are middle-income countries (MICs) [10], most VE studies to date have been conducted in high-income countries (HICs) [11]. Differences in population-level demographics and comorbidities in MICs compared to HICs could potentially lead to differences in overall vaccine performance. In addition, MICs in the WHO European Region

have used a broader variety of COVID -19 vaccine products compared to HICs, including inactivated virus vaccines [12]. In the WHO European Region, primary series COVID -19 vaccine coverage has been considerably lower in MICs compared to HICs; as of 4 August 2023, primary series COVID -19 vaccine coverage was 73.8% and booster dose coverage was 48.4% in HICs compared to 55.3% and 15.9%, respectively, in upper MICs [13]. Having local data on COVID-19 VE from MICs in the WHO European Region could provide the additional advantage of promoting increased vaccine uptake.

In Georgia, an upper-middle income country of 3.7 million people, the COVID-19 vaccine rollout began on 15 March 2021 [14]. For the initial vaccine rollout, both ChAdOx1-S vaccine (Oxford/AstraZeneca) and BNT162b2 vaccine (Pfizer-BioNTech) were procured via the COVAX facility mechanism. Later in 2021, BBIBP-CorV vaccine (Sinopharm), CoronaVac vaccine (Sinovac Life Sciences, Beijing, China), and additional BNT162b2 vaccines were procured independently by the government of Georgia. HCWs were among the initial priority groups for vaccination. In order to understand COVID-19 VE among HCWs in Georgia, we conducted a prospective cohort study of COVID-19 VE against symptomatic SARS-CoV-2 infection among HCWs in six hospitals in Georgia. Here we describe the results of the initial interim analysis (March–December, 2021).

## Materials and methods

### Study design

We conducted a prospective cohort study to evaluate VE against symptomatic SARS-CoV-2 infection among HCWs in six hospitals in Georgia. The study design was based on a VE guidance document published by the WHO Regional Office for Europe [15], and the study was conducted within the framework of WHO's Unity platform [16].

### Data collection and management

From 19 March–16 July 2021, we invited all HCWs ≥ 18 years old who were employed at the study hospitals and eligible to receive the COVID-19 vaccine to participate in the study, as previously described [14]. At the time of enrolment, contra-indications to the COVID-19 vaccine in Georgia included having had a SARS-CoV-2 infection in the previous 120 days, and having an acute febrile illness at the time of intended vaccination. HCWs could participate in the study regardless of their hospital role, prior infection status, or their intention to receive COVID-19 vaccine. We excluded participants who were not eligible for vaccination before their enrolment in the study due to a SARS-CoV-2infection in the previous 120 days, and we excluded participants with unknown vaccination status at the time of enrollment. All HCWs in the study received COVID-19 vaccine through the national vaccine campaign led by the Georgia Ministry of Health.

At enrolment, participants completed a questionnaire that included questions about socio-demographics, comorbidities, occupation, self-perceived health status, prior SARS-CoV-2 infection, and COVID-19 and influenza vaccination history. In addition, at enrolment, every participant provided a blood sample for serology testing.

Following enrolment, participants completed a weekly symptom questionnaire, administered by study personnel; participants who reported any symptoms included in the Georgia MoH suspected COVID-19 case definition (fever, cough, general weakness, fatigue, headache, muscle aches, sore throat, runny nose, shortness of breath, lack of appetite, nausea, vomiting, diarrhea, altered mental status, loss of taste, or loss of smell) provided a respiratory specimen, which was tested for SARS-CoV-2 by either RT-PCR or rapid antigen test (RAT), depending

on availability at each facility. In addition, during the entire study period, HCWs at all six hospitals could be tested routinely for SARS-CoV-2 by RT-PCR or RATs.

RT-PCR-positive samples were sent to the Richard G. Lugar Center for Public Health Research in Tbilisi, Georgia, where they underwent whole genome sequencing. Study staff verified participants' vaccination status through the National Immunization Registry and confirmed RT-PCR and RAT results through the national SARS-CoV-2 laboratory database. Participants who tested positive for SARS-CoV-2 by RT- PCR or RAT were administered a follow-up questionnaire 30 days after their positive test, which included additional questions about symptoms and medical care. All study data were entered securely and stored in REDCap [17].

## Serology

Phlebotomists collected serology specimens from participants at enrolment, and three and six months after enrolment. Serology samples were tested for anti-nucleocapsid antibodies and anti-spike antibodies by Roche Elecsys Anti-SARS-CoV-2 S immunoassay electrochemiluminescence immunoassay (ECLIA) [18]. For both serological assays, cutoff values were determined according to instructions from the package insert. Serology test results were used to identify previous COVID-19 infections, and to identify new infections in a sensitivity analysis that included only unvaccinated HCWs and HCWs who had received mRNA vaccines, as described in the vaccine effectiveness analysis section below.

## Sample size estimations

We conducted sample size estimations prior to enrolment to ensure robust estimates for the primary study objective, COVID -19 VE against symptomatic SARS-CoV-2 infection. We estimated COVID-19 vaccination coverage among HWs in Georgia of 60–90% and a varying incidence of SARS-CoV-2 infection among unvaccinated participants of 0.05–0.2 during a one-year study in order to detect VE between 50–90%. To allow for different scenarios (changes in infection rates in the community, different vaccines), analysis of secondary endpoints, and to account for likely drop-out rate of roughly 15%, 1600 participants were targeted for enrolment in the study. We did not undertake purposive sampling; all eligible HCWs were offered enrolment.

## Vaccine effectiveness analysis

For our primary outcome, because RAT testing occurred frequently and was not always accompanied by a PCR test, we measured primary series VE against a combined outcome of PCR-confirmed SARS-CoV-2 infection and/or RAT-confirmed COVID-19, which we defined as a positive test result in a participant who had symptom onset from 14 days before until four days after the date of specimen collection. Participants were considered fully vaccinated with their primary series ≥14 days after receipt of their second COVID-19 vaccine.

We conducted VE analyses for all primary series vaccination and then separately for primary series BNT162b2 and primary series BBIBP-CorV against outcomes of symptomatic infection, medically attended infection, and hospitalization. We also evaluated VE separately for the period in which SARS-CoV-2 B.1.617.2 (Delta variant) was predominant (5 July 2021 -– 5 December 2021), which we defined using sequencing data from study samples along with publicly available data from Georgia in the Global Initiative on Sharing All Influenza Data (GISAID) [19].

We also evaluated the combined effect of prior SARS-CoV-2 infection and COVID-19 vaccination on VE, using unvaccinated participants without prior infection as the category of reference. For the primary analysis, participants were considered to have had prior SARS-CoV-2

infection if they reported previous PCR-confirmed and/or RAT-confirmed infection and/or were seropositive for anti-nucleocapsid antibody at enrolment. For participants who received inactivated virus vaccine more than 5 days prior to enrolment, we defined prior infection by PCR and RAT only.

In addition, we measured primary series VE against both symptomatic and asymptomatic SARS-CoV-2 infection, measured by a combined endpoint of a PCR-positive test, a RAT-positive test, and/or seroconversion, which we defined as a positive three-month or six-month anti-nucleocapsid antibody test in a participant who was previously seronegative. For participants who seroconverted during the study but did not have symptomatic illness prior to their seroconversion, we estimated the time of asymptomatic infection as halfway between the last negative serological test and the subsequent positive serological test, taking into account a 3-week lag for seroconversion among asymptomatic persons [20]. For participants who had a symptomatic illness prior to their seroconversion, but did not have a positive PCR or RAT test, we assumed that the infection occurred on the date of onset of their symptomatic illness. For this analysis we only evaluated BNT162b2 vaccine effectiveness; we excluded participants who received BBIBP-CorV, as inactivated vaccines can generate the production of anti-nucleocapsid antibodies.

## Further analyses and sensitivity analyses

We examined waning immunity by comparing VE in the periods from 14–89 days, 90–179 days and ≥180 days since the second vaccine dose, for both the overall period and the period of Delta circulation only.

## Statistical model

VE was estimated as (1 –adjusted hazard ratio)*100. Hazard ratios (HRs) comparing vaccinated and unvaccinated were estimated using Cox proportional hazards models with vaccination as a time-varying exposure (vaccination status of some individuals changed over time from unvaccinated to vaccinated, and from one to two doses), which allowed participants to contribute person-time to more than one exposure category. Calendar time was used as the underlying time in the Cox regression.

We calculated unadjusted HR and used adjusted HR to estimate VE. Both estimates included hospital as a fixed effect. We adjusted the multivariable regression model using *a priori* fixed covariates (hospital, age, sex, prior SARS-CoV-2 infection) and considered potential confounders (role in hospital, hands-on care, face-to-face patient contact, smoking, household size, any comorbidity, body mass index [BMI] category, and self-perceived health status) that changed the VE estimate by more than 5%, according to the change-in-estimate approach. We used stratification to address violations of the proportional hazards assumption. All results presented in the results section reflect adjusted VE.

Participants with a SARS-CoV-2 infection prior to enrollment started contributing person-time when they were considered "at risk" of reinfection, which we defined as 90 days after their most recent positive PCR or RAT test or, for participants without who were seropositive at enrolment but did not have a history of a PCR- or RAT-positive test, four weeks after their positive enrolment serology.

Participants contributed person-time from enrolment, or from the start of time at risk for those with prior SARS-CoV-2 infection, until whichever of the following endpoints came first: 1) the day of the first SARS-CoV-2 infection, 2) the day of receipt of a third vaccine dose, or 3) the day of withdrawal from the study, or censor date for the analysis period (5 December

2021). Person-time of persons vaccinated with only one dose was excluded from all analyses from the day they received their first dose.

For all of the proposed analyses above, we only considered results from models that had achieved convergence and had a minimum of 5 events per category of vaccine status, ensuring our ability to construct reliable models.

### Ethics and study registration

The study was approved by the NCDC and WHO Research Ethics Review Committees (reference numbers IRB 2021–014 and CERC.0097C, respectively). The CDC humans review determined the activity to be a public health evaluation. All participants provided written informed consent. The study is also registered at clinicaltrials.gov (Identifier NCT04868448).

## Results

Of 3,849 HCWs working in the six hospitals, we enrolled 1592 (41%) participants, of whom 31 were excluded from the analysis for various reasons (Fig 1). Of the 1,561 HCWs included in the analysis, the median age was 40 (IQR: 30–53) and 1318 (84%) were female. Overall, 390 (25%) participants reported having at least one underlying chronic condition. Most HCWs were nurses or midwives [604 (39%)] and physicians [306 (20%)] (Table 1). Overall, 816 (52%) participants reported providing "hands-on care" to patients. Age and sex distribution, comorbid conditions, and occupation were similar across sites. Of the 1475 participants who were either unvaccinated at enrolment or received their first COVID-19 dose no more than 5 days prior to enrolment, 937 (63%) had evidence of prior infection by seropositivity for either anti-spike antibodies or anti-nucleocapsid antibodies.

At enrolment, 224 (14%) participants had already received one dose of COVID-19 vaccine, 37 (2%) participants had received two doses, and 1,300 (83%) participants were unvaccinated (Table 1). At the end of the follow-up period, 1082 (69%) participants had received two doses, and 479 (31%) participants remained unvaccinated (S1 Table in S1 File). Of the 1,082 participants who had received two vaccine doses by the end of the follow-up period, most [745 (69%)] received BNT162b2, while 238 (22%) received BBIBP-CorV, 58 (5%) received Corona-Vac, 26 (2%) received ChAdOx1-S and 14 (1%) received heterologous vaccination (S1 Table in S1 File and Fig 2). At the end of the analysis period, compared to participants who had received two doses of vaccine, unvaccinated participants were more likely to be doctors (24% vs. 10%) and were slightly younger (38 years (IQR = 28–52) vs. 41 years (IQR = 30–53)) but otherwise demographic, health, and occupation characteristics were similar between vaccinated and unvaccinated participants (S1 Table in S1 File).

During the study period, specimens were collected and tested from 686/796 (86%) reported symptomatic events. There were 124 symptomatic SARS-CoV-2 infections among unvaccinated participants (90 by PCR and 34 by RAT), and 67 symptomatic infections (52 by PCR and 15 by RAT) among participants who had completed a primary series (Table 2). Of the 191 symptomatic infections that occurred during the study period, 163 (85%) occurred between 1 July– 5 December 2021, a period during which the Delta variant predominated (Fig 3 and S1 Fig).

All participants who were infected during the study period completed the 30-day follow up questionnaire after their positive PCR or RAT test; 132/191 (69%) sought medical care, 26/188 (14%) sought care at an emergency department, 44/191 (23%) were hospitalized, 12/191 were admitted to the ICU and no one died.

Primary series VE against the combined outcome of symptomatic PCR-confirmed and RAT-confirmed infections was 58% (95% CI: 41, 70). Among participants without prior

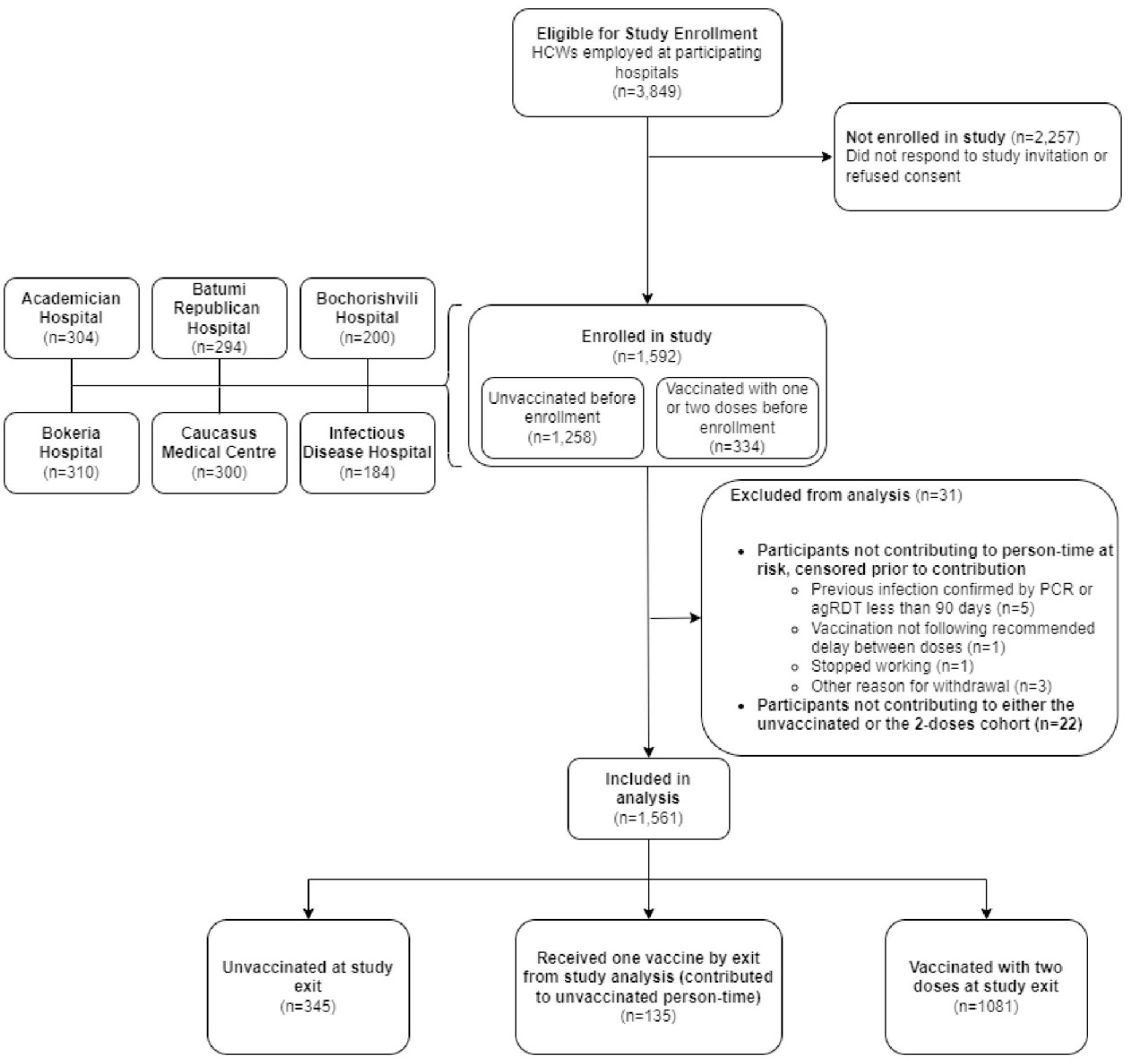

**Fig 1. Flowchart illustrating the enrolment of healthcare workers in COVID-19 vaccine effectiveness study, Georgia, 2021.**

infection, VE was 56% (95% CI: 35, 70), and among those with prior SARS-CoV-2 infection, VE was 58% (95% CI: 11, 80) (Table 2). For participants who received primary series BNT162b2 vaccine, overall VE was 68% (95% CI: 51, 79). In those without prior infection, primary series BNT162b2 VE was 63% (95% CI: 43, 77) (Table 2 and Fig 4). For participants who received primary series BBIBP-CorV vaccine, overall VE was 40% (95% CI: 1, 64). In those without prior infection, VE for primary series BBIBP-CorV was 31% (95%CI: -26, 62) (Table 2).

Overall VE against medically attended PCR- and RAT-confirmed COVID-19 was 52% (95% CI: 28, 67), and VE against PCR- and RAT-confirmed hospitalizations was 69% (95% CI: 36; 85) (Table 3 and Fig 4). During the Delta-predominant period, VE against medically attended COVID-19 was 41% (95% CI: 10, 61), and VE against hospitalization was 61% (95%

**Table 1. Demographic, occupational and health characteristics, prior infection status of participants in COVID-19 vaccine effectiveness study, by vaccination status at enrolment, Georgia, 2021.**

| Characteristic/Category | All Participants, n = 1561 | Unvaccinated, n = 1300 | Partially vaccinated (1 dose), n = 224 | Vaccinated with primary series (2 doses), n = 37 |
|---|---|---|---|---|
| **Age, n = 1561** | | | | |
| Median (IQR) | 40 (30–53) | 40 (28.8–52) | 47 (35–56.2) | 40 (31–52) |
| **Age group, n = 1561** | | | | |
| <20, n (%) | 16 (1) | 15 (1) | 1 (<1) | 0 (0) |
| 20–29, n (%) | 373 (24) | 335 (26) | 31 (14) | 7 (19) |
| 30–39, n (%) | 343 (22) | 291 (22) | 42 (19) | 10 (27) |
| 40–49, n (%) | 330 (21) | 270 (21) | 51 (23) | 9 (24) |
| 50–59, n (%) | 314 (20) | 248 (19) | 60 (27) | 6 (16) |
| 60+, n (%) | 185 (12) | 141 (11) | 39 (17) | 5 (14) |
| **Sex, n = 1561** | | | | |
| female, n (%) | 1318 (84) | 1105 (85) | 189 (84) | 24 (65) |
| male, n (%) | 243 (16) | 195 (15) | 35 (16) | 13 (35) |
| **Hospital, n = 1561** | | | | |
| Acad. K Central University Hosp., n (%) | 300 (19) | 240 (18) | 49 (22) | 11 (30) |
| Batumi Republican Hospital, n (%) | 276 (18) | 216 (17) | 52 (23) | 8 (22) |
| Bochorishvili Clinic, n (%) | 194 (12) | 178 (14) | 14 (6) | 2 (5) |
| Bokeria Tbilisi Referral Hospital, n (%) | 309 (20) | 266 (20) | 37 (17) | 6 (16) |
| Caucasus Medical Centre, n (%) | 299 (19) | 251 (19) | 40 (18) | 8 (22) |
| Infectious Disease Hospital, n (%) | 183 (12) | 149 (11) | 32 (14) | 2 (5) |
| **Occupation/Role in hospital, n = 1561** | | | | |
| Nurse or Midwife, n (%) | 604 (39) | 537 (41) | 58 (26) | 9 (24) |
| Medical Doctor, n (%) | 306 (20) | 181 (14) | 108 (48) | 17 (46) |
| Other, n (%) | 651 (42) | 582 (45) | 58 (26) | 11 (30) |
| **Household size, n = 1561** | | | | |
| 1–3, n (%) | 703 (45) | 579 (45) | 107 (48) | 17 (46) |
| 4–5, n (%) | 622 (40) | 526 (40) | 83 (37) | 13 (35) |
| 6+, n (%) | 236 (15) | 195 (15) | 34 (15) | 7 (19) |
| **Any chronic condition, n = 1561** | | | | |
| No, n (%) | 1171 (75) | 988 (76) | 155 (69) | 28 (76) |
| Yes, n (%) | 390 (25) | 312 (24) | 69 (31) | 9 (24) |
| **Number of chronic conditions, n = 1561** | | | | |
| 0, n (%) | 1171 (75) | 988 (76) | 155 (69) | 28 (76) |
| 1, n (%) | 307 (20) | 252 (19) | 48 (21) | 7 (19) |
| ≥2, n (%) | 83 (5) | 60 (5) | 21 (9) | 2 (5) |
| **Body mass index, n = 1561** | | | | |
| Underweight or normal, n (%) | 721 (46) | 607 (47) | 101 (45) | 13 (35) |
| Overweight, n (%) | 481 (31) | 394 (30) | 74 (33) | 13 (35) |
| Obese, n (%) | 359 (23) | 299 (23) | 49 (22) | 11 (30) |
| **Smoking, n = 1560** | | | | |
| Currently smokes, n (%) | 388 (25) | 323 (25) | 48 (21) | 17 (46) |
| Never smokes, n (%) | 1030 (66) | 865 (67) | 148 (66) | 17 (46) |
| Previously smokes, n (%) | 142 (9) | 111 (9) | 28 (12) | 3 (8) |
| **Self-assessed health status, n = 1561** | | | | |
| Excellent, n (%) | 127 (8) | 104 (8) | 17 (8) | 6 (16) |

*(Continued)*

**Table 1.** (Continued)

| Characteristic/Category | All Participants, n = 1561 | Unvaccinated, n = 1300 | Partially vaccinated (1 dose), n = 224 | Vaccinated with primary series (2 doses), n = 37 |
|---|---|---|---|---|
| Very good, n (%) | 252 (16) | 206 (16) | 37 (17) | 9 (24) |
| Good, n (%) | 521 (33) | 438 (34) | 68 (30) | 15 (41) |
| Fair, n (%) | 641 (41) | 533 (41) | 101 (45) | 7 (19) |
| Poor, n (%) | 20 (1) | 19 (1) | 1 (<1) | 0 (0) |
| **Hands on care, n = 1561** | | | | |
| No, n (%) | 745 (48) | 639 (49) | 91 (41) | 15 (41) |
| Yes, n (%) | 816 (52) | 661 (51) | 133 (59) | 22 (59) |
| **Received influenza vaccine during 2020–2021 influenza season, n = 1561** | | | | |
| No, n (%) | 1068 (68) | 944 (73) | 105 (47) | 19 (51) |
| Yes, n (%) | 492 (32) | 355 (27) | 119 (53) | 18 (49) |
| **Face-to-face patient contact, n = 1561** | | | | |
| No, n (%) | 336 (22) | 292 (22) | 37 (17) | 7 (19) |
| Yes, n (%) | 1125 (78) | 1008 (78) | 187 (81) | 30 (81) |
| **Previous SARS-CoV-2 infection (before enrollment) confirmed by PCR or RAT, n = 1561** | | | | |
| 0, n (%) | 814 (52) | 645 (50) | 136 (61) | 33 (89) |
| 1, n (%) | 747 (48) | 655 (50) | 88 (39) | 4 (11) |
| **Previous SARS-CoV-2 infection (before enrollment) confirmed by any test: PCR, RAT or serology, n = 1561** | | | | |
| 0, n (%) | 558 (36) | 437 (34) | 120 (54) | 1 (3) |
| 1, n (%) | 1003 (64) | 863 (66) | 104 (46) | 36 (97) |
| **Seropositive at enrolment (AntiS+ or AntiN+), n = 1555** | | | | |
| 0, n (%) | 569 (37) | 446 (34) | 122 (55) | 1 (3) |
| 1, n (%) | 986 (63) | 850 (66) | 100 (45) | 36 (97) |
| **Anti-S+, n = 1558** | | | | |
| 0, n (%) | 589 (38) | 460 (35) | 124 (56) | 5 (14) |
| 1, n (%) | 969 (62) | 838 (65) | 99 (44) | 32 (86) |
| **Anti-N+, n = 1552** | | | | |
| 0, n (%) | 639 (41) | 502 (39) | 130 (59) | 7 (19) |
| 1, n (%) | 913 (59) | 791 (61) | 92 (41) | 30 (81) |
| **Delay between first dose and start of person-time contribution, in days (n = 261)** | | | | |
| Median (IQR) | 2 (1–4) | _ | 2 (1–3) | 30 (29–31) |
| **Delay between second dose and start of person-time contribution, in days (n = 37)** | | | | |
| )Median (IQR) | 7 (3–9) | _ | _ | 7 (3–9) |
| **COVID-19 Vaccine product received prior to start of person-time contribution, n = 1561** | | | | |
| Unvaccinated, n (%) | 1300 (83) | 1300 (100) | 0 (0) | 0 (0) |
| ChAdOx1-S—2 doses, n (%) | 3 (<1) | 0 (0) | 0 (0) | 3 (8) |
| BNT162b2—1 dose, n (%) | 145 (9) | 0 (0) | 145 (65) | 0 (0) |
| BNT162b2—2 dose, n (%) | 26 (2) | 0 (0) | 0 (0) | 26 (70) |
| BBIBP-CorV—1 dose, n (%) | 33 (2) | 0 (0) | 33 (15) | 0 (0) |
| BBIBP-CorV—2 dose, n (%) | 4 (<1) | 0 (0) | 0 (0) | 4 (11) |
| CoronaVac—1 dose, n (%) | 28 (2) | 0 (0) | 28 (12) | 0 (0) |
| CoronaVac—2 doses, n (%) | 4 (<1) | 0 (0) | 0 (0) | 4 (11) |

CI: 13, 81). Overall, for BNT162b2, VE against medically attended COVID-19 was 63% (95% CI: 39; 78). Because of the low number of events, BNT162b2 VE against hospitalization could not be calculated, and BBIBP-CorV Ve against medically attended COVID-19 and hospitalization could not be calculated.

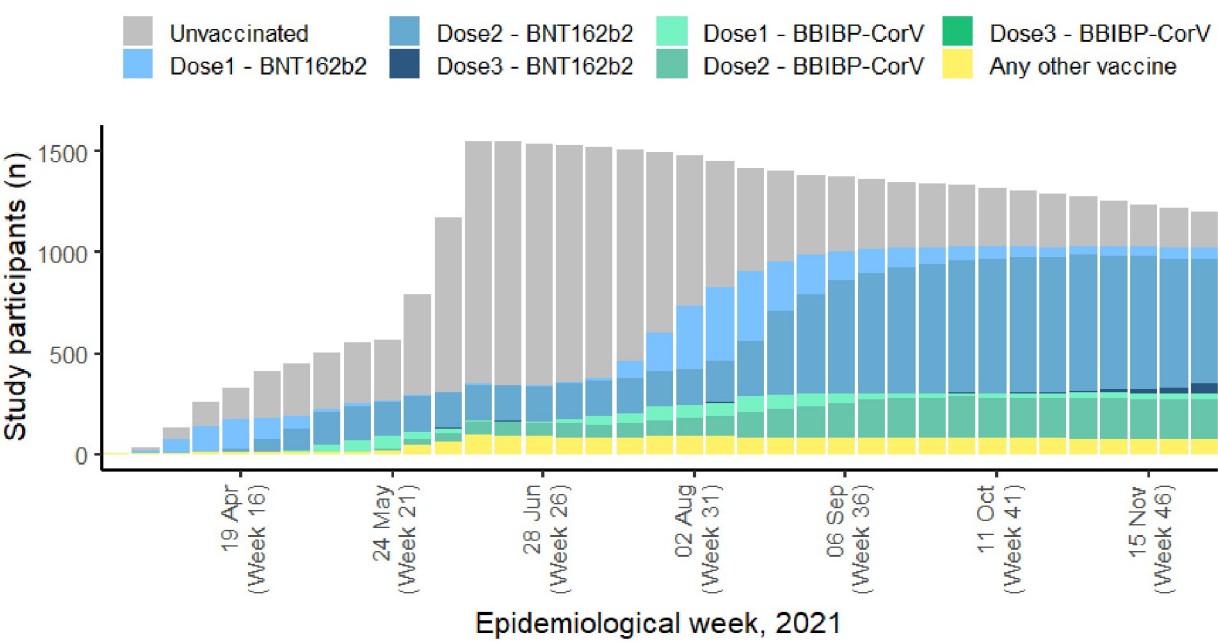

**Fig 2. COVID-19 vaccine coverage in the study population, by epidemiological week, Georgia, 2021.**

Primary series VE estimates against symptomatic, PCR-confirmed SARS-CoV-2 infection, medically attended infection, and hospitalization were similar to VE estimates for the combined PCR/RAT endpoint. (S2 Table in S1 File).

Compared to unvaccinated participants with no evidence of prior SARS-CoV-2 infection, VE against symptomatic PCR- and RAT-confirmed infection was 56% (95% CI: 35; 70) for vaccinated participants without evidence of prior infection, and 95% (95% CI: 90; –98) for vaccinated participants who had prior infection. Protection against symptomatic, PCR- and RAT-confirmed infection was 85% (95%CI: 77; –90) for unvaccinated participants who had been previously infected. Trends were similar when we limited this analysis to BNT162b2-only, BBIBP-CorV-only, and for all vaccines for the Delta period (Table 4).

In our evaluation of protection against the more inclusive outcome of infection confirmed by PCR, RAT, or seroconversion, which was limited to primary series BNT162b2 vaccination, we found similar trends. VE was 32% (95% CI: 0, 53) overall, 15% (95% CI: -31, –44) for those without previous infection, and 69% (95%CI: 26, 87) for those with previous infection (S3 Table in S1 File). Compared to unvaccinated participants who had not been previously infected, VE was 15% (95%CI: –31; 44) for primary series vaccination without previous infection, and 96% (95% CI: 92–98) for primary series vaccination among participants with prior infection. Unvaccinated participants who had been previous infected had 90% (95% CI: 87; 93) protection from re-infection (S4 Table in S1 File).

In our evaluation of VE against symptomatic PCR- and RAT-confirmed SARS-CoV-2 infection stratified by time since vaccination, for the overall analysis, (Table 5 and Fig 5), VE was 60% (95% CI: 39, –74) for participants 14–89 days after their second vaccine dose, 48% (95% CI: 18, 66) for participants who were 90–179 days after their second vaccine dose, and 14% (95%CI: -134, 68) for those ≥180 days. When we restricted the time-since-vaccination analysis to vaccine product and the Delta-predominant period, the trends were similar (Table 5. Fig 5).

Table 2. Vaccine effectiveness against symptomatic PCR and rapid antigen test confirmed SARS-CoV-2 infection for full cohort, and stratified by previous infection, vaccine brand and variant of interest.

| | | N participants | Total person-time (days) | PCR-confirmed symptomatic SARS-CoV-2 infection | RAT-confirmed symptomatic SARS-CoV-2 infection | All symptomatic SARS-CoV-2 infections | Unadjusted HR | (95% CI) | Unadjusted VE | (95%CI) | Adjusted VE | (95%CI) |
|---|---|---|---|---|---|---|---|---|---|---|---|---|
| Overall study period | **Two doses—any vaccine** | | | | | | | | | | | |
| | **Total cohort** | 1561 | | | | | | | | | | |
| | Unvaccinated | 1300 | 112050 | 90 | 34 | 124 | | | | | | |
| | ≥14d from 2nd dose | 1054 | 105080 | 52 | 15 | 67 | 0.56 | (0.40; 0.80) | 44 | (20; 60) | 58 | (41; 70) |
| | **Without Prior Infection** | 437 | 36109 | 63 | 25 | 88 | | | | | | |
| | Unvaccinated | | | | | | | | | | | |
| | ≥14d from 2nd dose | 357 | 37683 | 40 | 14 | 54 | 0.48 | (0.32; 0.70) | 52 | (30; 68) | 56 | (35; 70) |
| | **With Prior Infection** | 863 | 75941 | 27 | 9 | 36 | | | | | | |
| | Unvaccinated | | | | | | | | | | | |
| | ≥14d from 2nd dose | 697 | 67397 | 12 | 1 | 13 | 0.41 | (0.20; 0.84) | 59 | (16; 80) | 58 | (11; 80) |
| | **Two doses—BNT162b2 vaccine** | | | | | | | | | | | |
| | **Total cohort** | 1470 | | | | | | | | | | |
| | Unvaccinated | 1300 | 112050 | 90 | 34 | 124 | | | | | | |
| | ≥14d from 2nd dose | 732 | 72695 | 25 | 6 | 31 | 0.40 | (0.26; 0.63) | 60 | (37; 74) | 68 | (51; 79) |
| | **Without Prior Infection** | 437 | 36109 | 63 | 25 | 88 | | | | | | |
| | Unvaccinated | | | | | | | | | | | |
| | ≥14d from 2nd dose | 239 | 25763 | 22 | 6 | 28 | 0.37 | (0.24; 0.60) | 63 | (40; 76) | 63 | (43; 77) |
| | **Two doses—BBIBP-CorV vaccine** | | | | | | | | | | | |
| | **Total cohort** | 1337 | | | | | | | | | | |
| | Unvaccinated | 1300 | 112050 | 90 | 34 | 124 | | | | | | |
| | ≥14d from 2nd dose | 227 | 21136 | 18 | 6 | 24 | 0.87 | (0.52; 1.45) | 13 | (-45; 48) | 40 | (1; 64) |
| | **Without Prior Infection** | 437 | 36109 | 63 | 25 | 88 | | | | | | |
| | Unvaccinated | | | | | | | | | | | |
| | ≥14d from 2nd dose | 91 | 8977 | 15 | 5 | 20 | 0.74 | (0.41; 1.33) | 26 | (-33; 59) | 31 | (-26; 62) |

*(Continued)*

**Table 2.** (Continued)

| | | N participants | Total person-time (days) | PCR-confirmed symptomatic SARS-CoV-2 infection | RAT-confirmed symptomatic SARS-CoV-2 infection | All symptomatic SARS-CoV-2 infections | Unadjusted HR | (95% CI) | Unadjusted VE | (95%CI) | Adjusted VE | (95%CI) |
|---|---|---|---|---|---|---|---|---|---|---|---|---|
| **Delta period** | **Two doses—any vaccine** | | | | | | | | | | | |
| | Total cohort | 1556 | | | | | | | | | | |
| | Unvaccinated | 1162 | 72917 | 67 | 25 | 92 | | | | | | |
| | ≥14d from 2nd dose | 1068 | 96751 | 49 | 15 | 64 | 0.60 | (0.41; 0.86) | 40 | (14; 59) | 52 | (30; 66) |
| | **Without Prior Infection** | | | | | | | | | | | |
| | | 378 | 21896 | 45 | 19 | 64 | | | | | | |
| | Unvaccinated | | | | | | | | | | | |
| | ≥14d from 2nd dose | 339 | 30932 | 36 | 12 | 48 | 0.53 | (0.35; 0.80) | 47 | (20; 65) | 52 | (26; 69) |
| | **With Prior Infection** | | | | | | | | | | | |
| | Unvaccinated | 784 | 51021 | 22 | 6 | 28 | | | | | | |
| | ≥14d from 2nd dose | 729 | 65819 | 13 | 3 | 16 | 0.49 | (0.25; 0.98) | 51 | (2; 75) | 47 | (-10; 74) |
| | **Two doses—BNT162b2 vaccine** | | | | | | | | | | | |
| | Total cohort | 1371 | | | | | | | | | | |
| | Unvaccinated | 1162 | 72917 | 67 | 25 | 92 | | | | | | |
| | ≥14d from 2nd dose | 733 | 64109 | 23 | 6 | 29 | 0.46 | (0.28; 0.73) | 54 | (27; 72) | 61 | (38; 75) |
| | **Without Prior Infection** | | | | | | | | | | | |
| | | 378 | 21896 | 45 | 19 | 64 | | | | | | |
| | Unvaccinated | | | | | | | | | | | |
| | ≥14d from 2nd dose | 222 | 19615 | 19 | 5 | 24 | 0.46 | (0.28; 0.77) | 54 | (23; 72) | 55 | (26; 73) |
| | **Two doses—BBIBP-CorV vaccine** | | | | | | | | | | | |
| | Total cohort | 1250 | | | | | | | | | | |
| | Unvaccinated | 1161 | 72906 | 67 | 25 | 92 | | | | | | |
| | ≥14d from 2nd dose | 227 | 20221 | 18 | 6 | 24 | 0.88 | (0.52; 1.5) | 12 | (-50; 48) | 37 | (-9; 63) |
| | **Without Prior Infection** | | | | | | | | | | | |
| | | 378 | 21896 | 45 | 19 | 64 | | | | | | |
| | Unvaccinated | | | | | | | | | | | |
| | ≥14d from 2nd dose | 91 | 8474 | 15 | 5 | 20 | 0.72 | (0.39; 1.31) | 28 | (-31; 61) | 34 | (-23; 64) |

* due to the small number of events, brand-specific VE in participants with previous infection could not be estimated

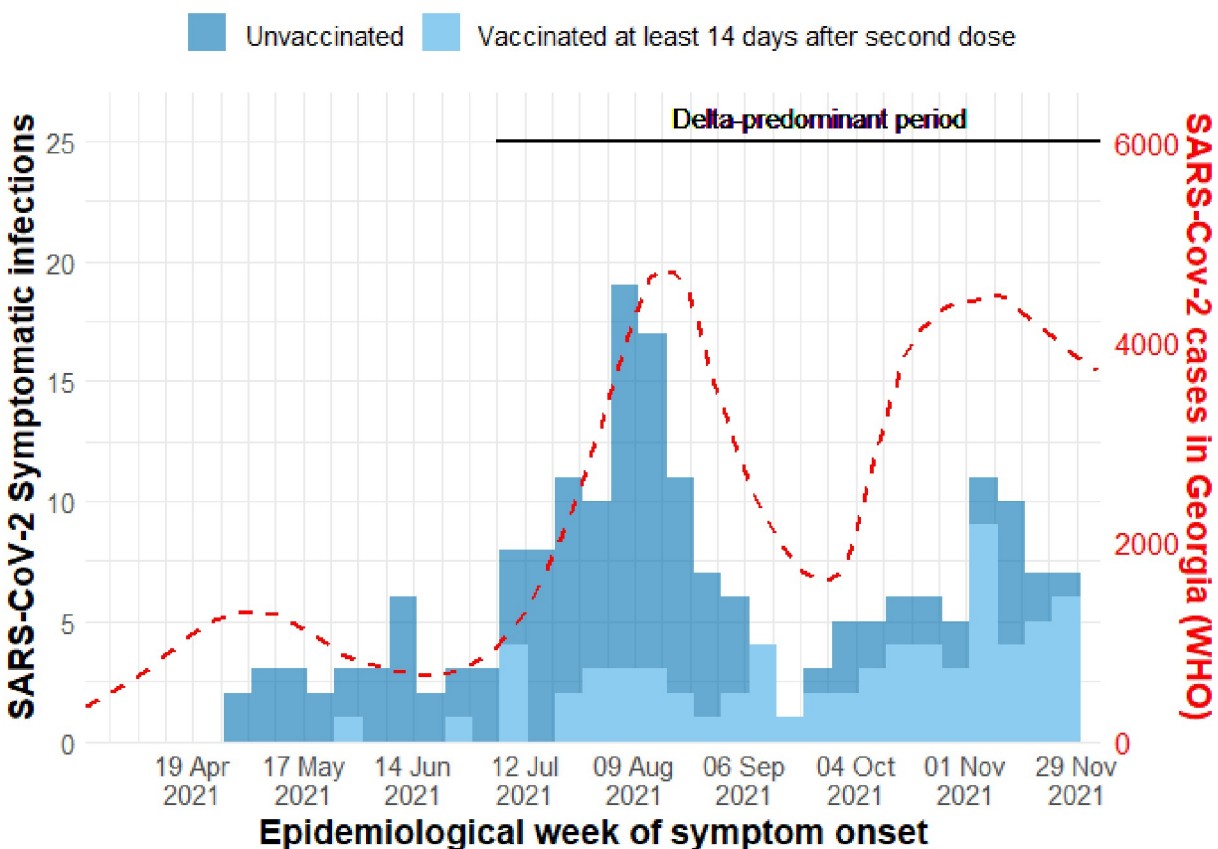

**Fig 3. Number of symptomatic COVID-19 cases by vaccination status in the study population and national COVID-19 incidence in Georgia, by epidemiologic week, 2021.**

S5 Table in S1 File presents the identified confounders for each model.

## Discussion

We found that primary series COVID-19 vaccination was nearly 60% effective in preventing symptomatic SARS-CoV-2 illness among Georgian HCWs, most of whom had been previously infected with SARS-CoV-2, during a period of mostly Delta variant circulation. Both primary series BNT162b2 vaccination and primary series BBIBP-CorV vaccination conferred similarly high VE against symptomatic infection. Our findings support the current recommendations of Georgia that HCWs, and all adults over 18 years old, should receive primary series COVID-19 vaccination. The results from this study, which to our knowledge is the first study to describe COVID-19 VE in Georgia, could be used to promote increased COVID-19 vaccination in the country, which has one of the lowest COVID-19 vaccination rates in the WHO European region; as of the week of 4 June 2023 only 57% of Georgian HCWs had received primary series COVID-19 vaccination, and only 18% had received a booster dose. As of the same date, only 32% of the general population in Georgia had received primary series COVID-19 vaccination, and only 6% had received a booster dose [12].

Our findings of primary series BNT162b2 VE against symptomatic SARS-CoV-2 infection are similar to other previously published studies that evaluated VE against COVID-19 during periods of Delta circulation. A living active VE literature review and associated systematic

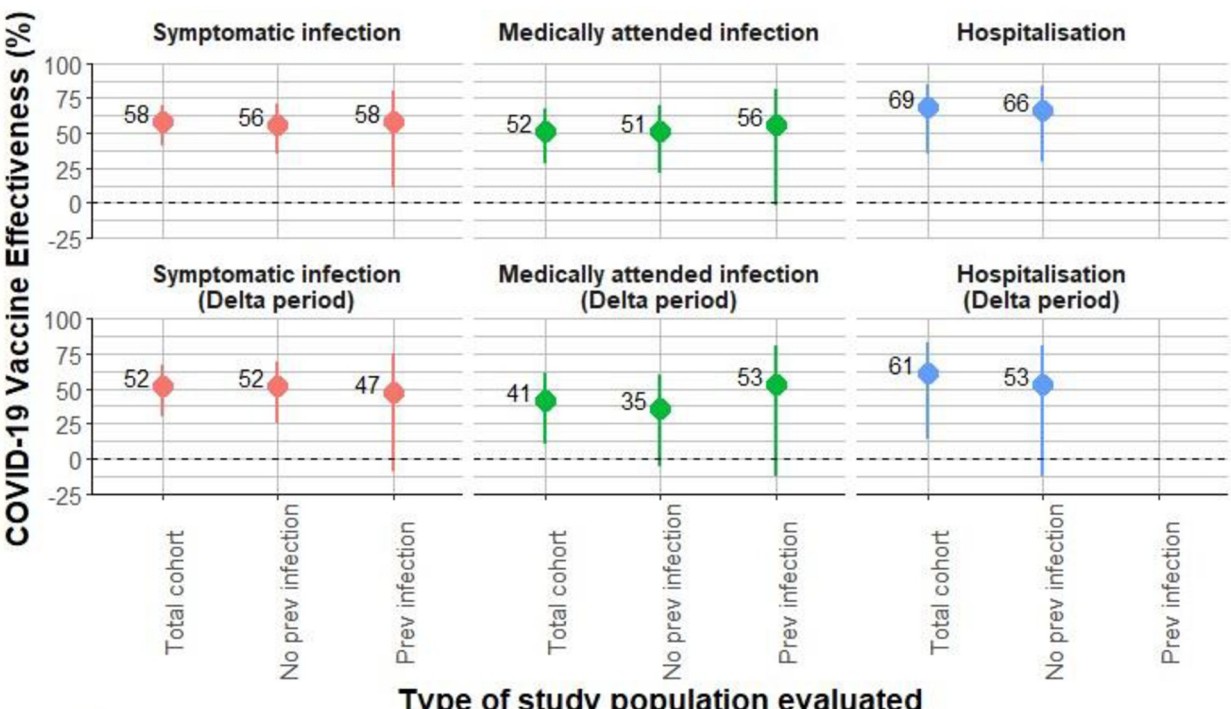

**Fig 4. COVID-19 vaccine effectiveness against symptomatic infection and medically attended cohort for total study cohort, and stratified by previous infection status, for overall study period, and for Delta-predominant period, Georgia, 2021.**

review [21] described point estimates for studies that evaluated primary series BNT162b2 VE against symptomatic disease during Delta ranging from 80–95% in the 14 days to <3 months following completion of the primary series; VE dropped to 45–80% after 3–6 months [11]. Studies from other regions of the world have demonstrated a consistently relatively high VE (70%) of primary series BNT162b2 against severe disease, which has increased following booster doses [11].

Although well over 1.5 billion doses of BBIBP-CorV have been used globally [22], very few post-marketing studies have evaluated its effectiveness [11]. Our study is one of few studies globally to evaluate VE of primary series BBIBP-CorV against symptomatic infection during the Delta period [11]. Our finding of 40% VE was in the range of primary series BBIBP-CorV VE against symptomatic infection during Delta described in two studies from China, which found adjusted VE of 50% (95%CI: 4; –74) [23] and 75 (95%CI: 6; –94) [24], and one study from Egypt, that found VE of 67% (95%CI: 43; 80) [25]. Studies of BBIBP-CoRV against more severe outcomes have shown mixed results, but have consistently shown the added benefit of booster doses, mostly against Delta but also against Omicron; a study of primary series BBIBP-CorV VE against hospitalization among people aged 18–64 years old in Hungary during Delta found a VE of 54% (44; 62) in the 14–120 days following the second dose that decreased over time, but increased to 77–95% following homologous or heterologous booster [26]; for 65–100 year-olds, the same study found slightly lower VE against hospitalization with similar trends following booster doses. A study during a period of primarily Delta circulation in Iran found that primary series BBIBP-CorV VE peaked at 85% (95%CI: 77; 91%) against

**Table 3. Vaccine effectiveness against medically attended COVID-19 and COVID-19 hospitalization, for full cohort, and stratified by previous infection, vaccine product, for overall study period and for Delta-predominant period, Georgia, 2021.**

| | | N participants | Total person-time (days) | PCR-confirmed symptomatic SARS-CoV-2 infection | RAT-confirmed symptomatic SARS-CoV-2 infection | All symptomatic SARS-CoV-2 infections | Unadjusted HR | (95% CI) | Unadjusted VE | (95%CI) | Adjusted VE | (95%CI) |
|---|---|---|---|---|---|---|---|---|---|---|---|---|
| **Overall period** | **Medically attended SARS-CoV-2 infection** | | | | | | | | | | | |
| | **Two doses—any vaccine** | | | | | | | | | | | |
| | **Total cohort** | 1561 | | | | | | | | | | |
| | Unvaccinated | 1300 | 112050 | 64 | 22 | 86 | | | | | | |
| | ≥14d from 2nd dose | 1054 | 105080 | 37 | 9 | 46 | 0.64 | (0.43; 0.95) | 36 | (5; 57) | 52 | (28; 67) |
| | **Without Prior Infection** | | | | | | | | | | | |
| | Unvaccinated | 437 | 36109 | 47 | 15 | 62 | | | | | | |
| | ≥14d from 2nd dose | 357 | 37683 | 30 | 8 | 38 | 0.55 | (0.35; 0.85) | 45 | (15; 65) | 51 | (21; 70) |
| | **With Prior Infection** | | | | | | | | | | | |
| | Unvaccinated | 863 | 75941 | 17 | 7 | 24 | | | | | | |
| | ≥14d from 2nd dose | 697 | 67397 | 7 | 1 | 8 | 0.42 | (0.18; 0.98) | 58 | (2; 82) | 56 | (-2; 81) |
| | **Two doses—BNT162b2** | | | | | | | | | | | |
| | **Total cohort** | 1470 | | | | | | | | | | |
| | Unvaccinated | 1300 | 112050 | 64 | 22 | 86 | | | | | | |
| | ≥14d from 2nd dose | 732 | 72695 | 18 | 5 | 23 | 0.49 | (0.30; 0.81) | 51 | (19; 70) | 63 | (39; 78) |
| | **Without Prior Infection** | | | | | | | | | | | |
| | Unvaccinated | 437 | 36109 | 47 | 15 | 62 | | | | | | |
| | ≥14d from 2nd dose | 239 | 25763 | 16 | 5 | 21 | 0.44 | (0.26; 0.75) | 56 | (25; 74) | 54 | (24; 72) |
| **Delta period** | **Two doses—any vaccine** | | | | | | | | | | | |
| | **Total cohort** | 1556 | | | | | | | | | | |
| | Unvaccinated | 1162 | 72917 | 45 | 13 | 58 | | | | | | |
| | ≥14d from 2nd dose | 1068 | 96751 | 35 | 9 | 44 | 0.69 | (0.45; 1.08) | 31 | (-8; 55) | 41 | (10; 61) |
| | **Without Prior Infection** | | | | | | | | | | | |
| | Unvaccinated | 378 | 21896 | 31 | 9 | 40 | | | | | | |
| | ≥14d from 2nd dose | 339 | 30932 | 28 | 7 | 35 | 0.68 | (0.42; 1.11) | 32 | (-11; 58) | 35 | (-6; 60) |
| | **With Prior Infection** | | | | | | | | | | | |
| | Unvaccinated | 784 | 51021 | 14 | 4 | 18 | | | | | | |
| | ≥14d from 2nd dose | 729 | 65819 | 7 | 2 | 9 | 0.44 | (0.19; 1.02) | 56 | (-2; 81) | 53 | (-13; 80) |

*(Continued)*

**Table 3.** (Continued)

| | | N participants | Total person-time (days) | PCR-confirmed symptomatic SARS-CoV-2 infection | RAT-confirmed symptomatic SARS-CoV-2 infection | All symptomatic SARS-CoV-2 infections | Unadjusted HR | (95% CI) | Unadjusted VE | (95%CI) | Adjusted VE | (95%CI) |
|---|---|---|---|---|---|---|---|---|---|---|---|---|
| Overall period | **Hospitalization due to SARS-CoV-2 infection** | | | | | | | | | | | |
| | **Two doses—any vaccine** | | | | | | | | | | | |
| | **Total cohort** | 1561 | | | | | | | | | | |
| | Unvaccinated | 1300 | 112050 | 25 | 8 | 33 | | | | | | |
| | ≥14d from 2nd dose | 1054 | 105080 | 9 | 2 | 11 | 0.46 | (0.22; 0.96) | 54 | (4; 78) | 69 | (36; 85) |
| | **Without Prior Infection** | | | | | | | | | | | |
| | Unvaccinated | 437 | 36109 | 21 | 6 | 27 | | | | | | |
| | ≥14d from 2nd dose | 357 | 37683 | 8 | 2 | 10 | 0.37 | (0.18; 0.78) | 63 | (22; 82) | 66 | (29; 84) |
| Delta period | **Two doses—any vaccine** | | | | | | | | | | | |
| | **Total cohort** | 1556 | | | | | | | | | | |
| | Unvaccinated | 1162 | 72917 | 13 | 3 | 16 | | | | | | |
| | ≥14d from 2nd dose | 1068 | 96751 | 8 | 2 | 10 | 0.5 | (0.21; 1.17) | 50 | (-17; 79) | 61 | (13; 81) |
| | **Without Prior Infection** | | | | | | | | | | | |
| | Unvaccinated | 378 | 21896 | 10 | 2 | 12 | | | | | | |
| | ≥14d from 2nd dose | 339 | 30932 | 7 | 2 | 9 | 0.46 | (0.19; 1.11) | 54 | (-11; 81) | 53 | (-13; 81) |

* due to small number of events, brand specific VE in participants with previous infection could not be estimated

**Because of the low number of events, BNT162b2 vaccine effectiveness against hospitalization could not be calculated, and BBIBP-CorV vaccine effectiveness against medically attended COVID-19 and hospitalization could not be calculated.

hospitalization ≥ 151 days after receipt of the second dose and 56% (95% CI 33; 71%) against death 91–120 days after receipt of the second dose [27]. A study from Thailand found BBIBP--CorV primary series VE against pneumonia requiring invasive ventilation during Omicron of 66% (95%CI: 39; 81%) that had a non-statistically significant increase to 81% (95%CI: 36; 95%) following a booster dose [28].

In our study, nearly two-thirds of participants (64%) had been previously infected with SARS-CoV-2 at the time of their enrolment into the study. Nevertheless, we found clear benefit to COVID-19 vaccination; among participants with prior infection, primary series VE with any vaccine was 58%. The benefits of hybrid immunity–immunity conferred by the combination of vaccination and infection–have been widely described globally and within the WHO European region [29–31]. The added benefit of vaccination after infection is becoming an increasingly important public message as more of the world's population has experienced at least one SARS-CoV-2 infection [32].

Our findings of that COVID-19 vaccine prevented nearly two-thirds of symptomatic infections has positive implications for the role of vaccine in protecting the health of HCWs, the health of the patients and improving the resilience of the Georgian healthcare system. These findings underscore the importance of tailored messaging to highlight these points, and health policies to encourage increased COVID-19 vaccine uptake among HCWs.

**Table 4. Combined effect of previous infection and primary series vaccination in protecting against symptomatic PCR and/or RAT-confirmed SARS-CoV-2 infection, overall, by vaccine product, and for Delta-predominant period only, Georgia, 2021.**

| | | N participants | Total person-time (days) | PCR-confirmed symptomatic SARS-CoV-2 infection | RAT-confirmed symptomatic SARS-CoV-2 infection | All symptomatic SARS-CoV-2 infections | Unadjusted HR | (95% CI) | Unadjusted VE | (95%CI) | Adjusted VE | (95%CI) |
|---|---|---|---|---|---|---|---|---|---|---|---|---|
| Overall period | **Two doses (all vaccines)** | 1561 | | | | | | | | | | |
| | Unvaccinated without previous infection [ref] | 437 | 36109 | 63 | 25 | 88 | | | | | | |
| | Unvaccinated with previous infection | 863 | 75941 | 27 | 9 | 36 | 0.17 | (0.11; 0.25) | 83 | (75; 89) | 85 | (77; 90) |
| | ≥14d from 2nd dose without previous infection | 357 | 37683 | 40 | 14 | 54 | 0.48 | (0.32; 0.70) | 52 | (30; 68) | 56 | (35; 70) |
| | ≥14d from 2nd dose with previous infection | 697 | 67397 | 12 | 1 | 13 | 0.06 | (0.03; 0.13) | 94 | (87; 97) | 95 | (90; 98) |
| | **Two doses (BNT162b2)** | 1470 | | | | | | | | | | |
| | Unvaccinated without previous infection [ref] | 437 | 36109 | 63 | 25 | 88 | | | | | | |
| | Unvaccinated with previous infection | 863 | 75941 | 27 | 9 | 36 | 0.17 | (0.11; 0.25) | 83 | (75; 89) | 85 | (77; 90) |
| | ≥14d from 2nd dose without previous infection | 239 | 25763 | 22 | 6 | 28 | 0.37 | (0.24; 0.60) | 63 | (40; 76) | 64 | (43; 77) |
| | **Two doses (BBIBP-CorV)** | 1337 | | | | | | | | | | |
| | Unvaccinated without previous infection [ref] | 437 | 36109 | 63 | 25 | 88 | | | | | | |
| | Unvaccinated with previous infection | 863 | 75941 | 27 | 9 | 36 | 0.17 | (0.11; 0.25) | 83 | (75; 89) | 85 | (77; 90) |
| | ≥14d from 2nd dose without previous infection | 91 | 8977 | 15 | 5 | 20 | 0.74 | (0.41; 1.33) | 26 | (-33; 59) | 31 | (-26; 62) |

(Continued)

**Table 4.** (Continued)

| | N participants | Total person-time (days) | PCR-confirmed symptomatic SARS-CoV-2 infection | RAT-confirmed symptomatic SARS-CoV-2 infection | All symptomatic SARS-CoV-2 infections | Unadjusted HR | (95% CI) | Unadjusted VE | (95%CI) | Adjusted VE | (95%CI) |
|---|---|---|---|---|---|---|---|---|---|---|---|
| **Delta period** | | | | | | | | | | | |
| **Two doses (all vaccines)** | 1556 | | | | | | | | | | |
| Unvaccinated without previous infection [ref] | 378 | 21896 | 45 | 19 | 64 | | | | | | |
| Unvaccinated with previous infection | 784 | 51021 | 22 | 6 | 28 | 0.17 | (0.11; 0.27) | 83 | (73; 89) | 86 | (77; 91) |
| ≥14d from 2nd dose without previous infection | 339 | 30932 | 36 | 12 | 48 | 0.53 | (0.35; 0.80) | 47 | (20; 65) | 52 | (26; 69) |
| ≥14d from 2nd dose with previous infection | 729 | 65819 | 13 | 3 | 16 | 0.08 | (0.04; 0.16) | 92 | (84; 96) | 93 | (86; 96) |
| **Two doses (BNT162b2)/ Previous infection status** | 1371 | | | | | | | | | | |
| Unvaccinated without previous infection [ref] | 378 | 21896 | 45 | 19 | 64 | | | | | | |
| Unvaccinated with previous infection | 784 | 51021 | 22 | 6 | 28 | 0.17 | (0.11; 0.27) | 83 | (73; 89) | 86 | (77; 91) |
| ≥14d from 2nd dose without previous infection | 222 | 19615 | 19 | 5 | 24 | 0.46 | (0.28; 0.77) | 54 | (23; 72) | 55 | (26; 73) |
| **Two doses (BBIBP-CorV)** | 1250 | | | | | | | | | | |
| Unvaccinated without previous infection [ref] | 378 | 21896 | 45 | 19 | 64 | | | | | | |
| Unvaccinated with previous infection | 783 | 51010 | 22 | 6 | 28 | 0.17 | (0.11; 0.27) | 83 | (73; 89) | 86 | (77; 91) |
| ≥14d from 2nd dose without previous infection | 91 | 8474 | 15 | 5 | 20 | 0.72 | (0.39; 1.31) | 28 | (-31; 61) | 34 | (-23; 64) |

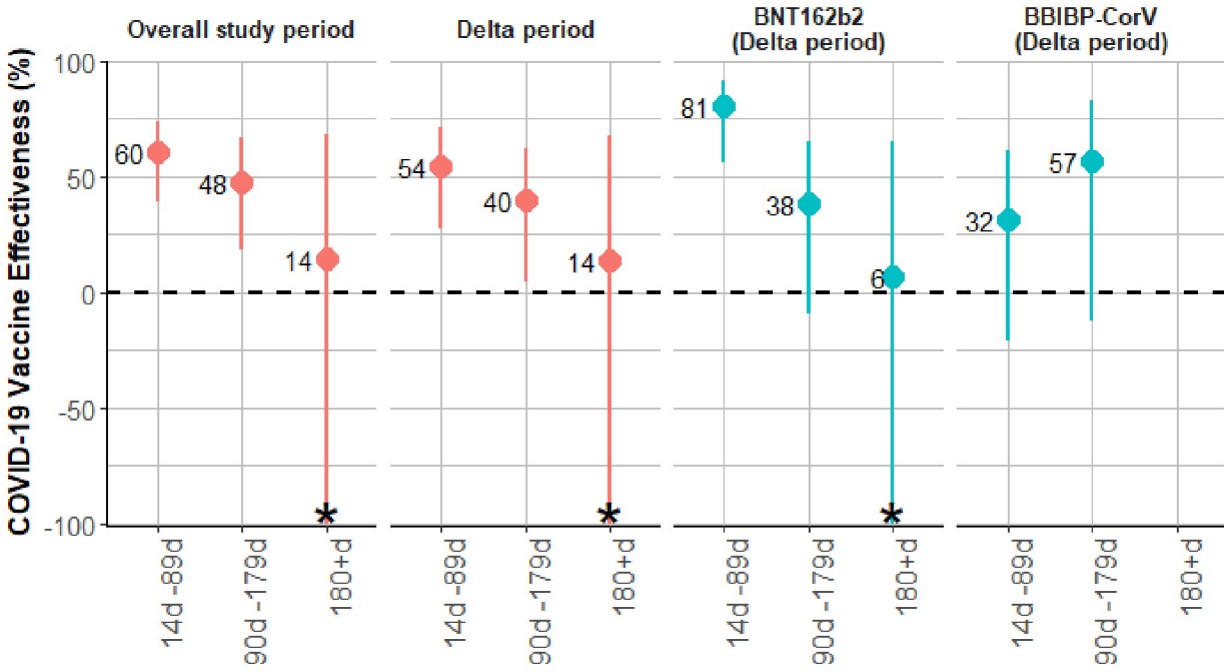

Note: *Confidence intervals for VE lower than -100% not printed in this graph

**Fig 5. COVID-19 primary series vaccine effectiveness by days since vaccination against symptomatic infection for overall cohort during entire study period, overall cohort during the Delta-predominant period, and for BNT162b2 and BBIBP CorV during the Delta-predominant period, Georgia, 2021.**

Our study had a number of strengths. By using PCR-confirmed infection and serology to define prior infection, and by using these two diagnostic tools and RAT to estimate VE against all infections, we were able to more comprehensively identify infections among participants in our study. In addition, by collecting quarterly serology samples, we were able to identify asymptomatic infections and symptomatic infections that may have been missed by PCR on one-time swab collection. Very few participants [3/1592 (0.2%)] withdrew from the study during the follow-up period. In addition, 85% of participants who reported a symptomatic illness on their weekly questionnaire had a specimen collected for PCR testing.

Our study also has limitations. First, while we evaluated COVID-19 VE against symptomatic and asymptomatic infections, endpoints that are particularly relevant for HCWs, who need to be healthy in order to provide clinical care and infection-free so that they do not pass on the virus to their patients, our study was not powered to estimate VE against severe outcomes such as hospitalization and death. In addition, our study evaluated primary series VE during a period of Delta circulation, and could not evaluate VE of booster doses; both will be evaluated in future analyses of this ongoing cohort study. Our study may suffer from selection bias; enrolment in the study was voluntary, HCWs in our study may not fully represent HCWs in Georgia. However, we did enroll over 40% of eligible HCWs in the six hospitals.

Additionally, while serology was a strength of our study in identifying previous infections and new infections (in those participants who were not vaccinated with inactivated virus vaccines), our anti-nucleocapsid antibody likely did not capture all previous infections, due to a combination of antibody waning, imperfect sensitivity of the assay, and potentially variable production of anti-nucleocapsid antibodies in vaccinated individuals [33]. Furthermore, we

**Table 5. Effect of time since vaccination on COVID-19 primary vaccine series effectiveness against symptomatic PCR- and RAT-confirmed infection, for full cohort, by vaccine product, and for Delta-predominant period, Georgia, 2021.**

| | | N participants | Total person-time (days) | PCR-confirmed symptomatic SARS-CoV-2 infection | RAT-confirmed symptomatic SARS-CoV-2 infection | All symptomatic SARS-CoV-2 infections | Unadjusted HR | (95% CI) | Unadjusted VE | (95% CI) | Adjusted VE | (95%CI) |
|---|---|---|---|---|---|---|---|---|---|---|---|---|
| Overall period | **Time since two doses (all vaccines)** | 1561 | | | | | | | | | | |
| | Unvaccinated [ref] | 1300 | 112050 | 90 | 34 | 124 | | | | | | |
| | 14d–89d from 2nd dose | 1054 | 71373 | 28 | 7 | 35 | 0.47 | (0.30; 0.72) | 53 | (28; 70) | 60 | (39; 74) |
| | 90d–179d from 2nd dose | 697 | 29150 | 19 | 7 | 26 | 0.73 | (0.46; 1.16) | 27 | (-16; 54) | 48 | (18; 66) |
| | ≥180d from 2nd dose | 186 | 4557 | 5 | 1 | 6 | 0.92 | (0.36; 2.38) | 8 | (-138; 64) | 14 | (-134; 68) |
| | **Time since two doses (BNT162b2)** | 1470 | | | | | | | | | | |
| | Unvaccinated [ref] | 1300 | 112050 | 90 | 34 | 124 | | | | | | |
| | 14d–89d from 2nd dose | 732 | 49837 | 9 | 1 | 10 | 0.21 | (0.1; 0.44) | 79 | (56; 90) | 82 | (63; 91) |
| | 90d–179d from 2nd dose | 485 | 18572 | 11 | 4 | 15 | 0.71 | (0.40; 1.24) | 29 | (-24; 60) | 52 | (13; 73) |
| | ≥180d from 2nd dose | 151 | 4286 | 5 | 1 | 6 | 0.95 | (0.37; 2.48) | 5 | (-148; 63) | 40 | (-55; 77) |
| | **Time since two doses (BBIBP-CorV)** | 1337 | | | | | | | | | | |
| | Unvaccinated [ref] | 1300 | 112050 | 90 | 34 | 124 | | | | | | |
| | 14d–89d from 2nd dose | 227 | 14894 | 13 | 5 | 18 | 0.85 | (0.49; 1.50) | 15 | (-50; 51) | 36 | (-10; 63) |
| | 90d–179d from 2nd dose | 132 | 6108 | 5 | 1 | 6 | 0.76 | (0.30; 1.91) | 24 | (-91; 70) | 57 | (-9; 83) |

(*Continued*)

**Table 5.** (Continued)

| | | N participants | Total person-time (days) | PCR-confirmed symptomatic SARS-CoV-2 infection | RAT-confirmed symptomatic SARS-CoV-2 infection | All symptomatic SARS-CoV-2 infections | Unadjusted HR | (95% CI) | Unadjusted VE | (95% CI) | Adjusted VE | (95%CI) |
|---|---|---|---|---|---|---|---|---|---|---|---|---|
| **Delta period** | **Time since two doses (all vaccines)** | 1556 | | | | | | | | | | |
| | Unvaccinated [ref] | 1162 | 72917 | 67 | 25 | 92 | | | | | | |
| | 14d–89d from 2nd dose | 1043 | 61860 | 25 | 7 | 32 | 0.47 | (0.30; 0.75) | 53 | (25; 70) | 54 | (28; 71) |
| | 90d–179d from 2nd dose | 715 | 30230 | 19 | 7 | 26 | 0.74 | (0.46; 1.21) | 26 | (-21; 54) | 40 | (4; 62) |
| | ≥180d from 2nd dose | 189 | 4661 | 5 | 1 | 6 | 1 | (0.38; 2.60) | -0 | (-160; 62) | 14 | (-130; 68) |
| | **Time since two doses (BNT162b2)** | 1371 | | | | | | | | | | |
| | Unvaccinated [ref] | 1162 | 72917 | 67 | 25 | 92 | | | | | | |
| | 14d–89d from 2nd dose | 708 | 41088 | 7 | 1 | 8 | 0.21 | (0.09; 0.49) | 79 | (51; 91) | 81 | (56; 91) |
| | 90d–179d from 2nd dose | 489 | 18631 | 11 | 4 | 15 | 0.77 | (0.44; 1.37) | 23 | (-37; 56) | 38 | (-9; 65) |
| | ≥180d from 2nd dose | 154 | 4390 | 5 | 1 | 6 | 1.03 | (0.39; 2.72) | -3 | (-172; 61) | 6 | (-151;65) |
| | **Time since two doses (BBIBP-CorV)** | 1250 | | | | | | | | | | |
| | Unvaccinated [ref] | 1161 | 72906 | 67 | 25 | 92 | | | | | | |
| | 14d–89d from 2nd dose | 227 | 13979 | 13 | 5 | 18 | 0.91 | (0.52; 1.59) | 9 | (-59; 48) | 32 | (-21; 61) |
| | 90d–179d from 2nd dose | 132 | 6108 | 5 | 1 | 6 | 0.75 | (0.30; 1.91) | 25 | (-91; 70) | 57 | (-12; 83) |

were not able to use serology test results to identify previous infections in 86 (5.5%) HCWs who had received their first COVID-19 vaccine more than five days prior to enrolment; however, we believe that by defining previous by a composite of documented PCR and RAT results, combined with serology test results, we captured the nearly all previous infections, and the amount of missed previous infections would be very unlikely to meaningfully impact our results. While the Georgian government encouraged routine testing of all HCWs during the study period, not all HCWs in our study were tested routinely with the same frequency, which may have introduced bias. In our study the unvaccinated group at the end of the study period was slightly younger and included more physicians; however, in order to address these differences, we controlled for age and occupation in our final adjusted models.

## Conclusions

In conclusion, we found that a primary series COVID-19 vaccination, which included mainly BNT162b2 vaccine and, to a lesser extent, BBIBP-CorV, was effective in preventing symptomatic infection in hospital-based HCWs in Georgia. Our findings support current vaccine policy and underscore the need to promote vaccine uptake in Georgia both in HCWs and the general population, where uptake has lagged in comparison to most other countries in the European region of WHO. Our findings also add to the growing literature on the added benefit of COVID-19 vaccination in individuals who have been previously infected with SARS-CoV-2.

## Supporting information

**S1 Fig. Whole genome sequencing results of samples from SARS-CoV-2 positive cases in Georgia by week during the study analysis period, 2021\*.**
(DOCX)

**S1 File.**
(DOCX)

## Acknowledgments

We are grateful to the study team at the Georgia National Centers for Disease Control and at all the participating study hospitals. In addition we would like to thank the following colleagues: Pernille Jorgensen and Diogo Simao Lemos (WHO Regional Office for Europe), Irina Begiashvilli (WHO Country Office in Georgia), Lindsey Duca (US CDC), Marta Valenciano and Alain Moren (Epiconcept). This study has been conducted within the framework of the WHO Strategic Preparedness and Response plan and WHO/European Regional Office Response for COVID-19.

**Disclaimer**: The authors affiliated with the World Health Organization (WHO) are alone responsible for the views expressed in this publication, and they do not necessarily represent the decisions or the policies of the WHO.

## Author Contributions

**Conceptualization:** Mark A. Katz, Madelyn Yiseth Rojas Castro, Giorgi Chakhunashvili, C. Jason McKnight, Héloïse Lucaccioni, Tamila Zardiashvili, Richard Pebody, Esther Kissling, Lia Sanodze.

**Data curation:** Madelyn Yiseth Rojas Castro, Giorgi Chakhunashvili, Caleb L. Ward, Iris Finci, Esther Kissling, Lia Sanodze.

**Formal analysis:** Madelyn Yiseth Rojas Castro, Esther Kissling.

**Funding acquisition:** Mark A. Katz, Tamila Zardiashvili, Richard Pebody, Lia Sanodze.

**Investigation:** Mark A. Katz, Giorgi Chakhunashvili, Nazibrola Chitadze, Caleb L. Ward, Héloïse Lucaccioni, Lia Sanodze.

**Methodology:** Mark A. Katz, Madelyn Yiseth Rojas Castro, Giorgi Chakhunashvili, Nazibrola Chitadze, C. Jason McKnight, Héloïse Lucaccioni, Iris Finci, Richard Pebody, Esther Kissling, Lia Sanodze.

**Project administration:** Mark A. Katz, Giorgi Chakhunashvili, Nazibrola Chitadze, Caleb L. Ward, Héloïse Lucaccioni, Iris Finci, Tamila Zardiashvili, Lia Sanodze.

**Resources:** Mark A. Katz, Lia Sanodze.

**Supervision:** Mark A. Katz, Giorgi Chakhunashvili, Iris Finci, Richard Pebody, Esther Kissling, Lia Sanodze.

**Validation:** Madelyn Yiseth Rojas Castro, Giorgi Chakhunashvili, Caleb L. Ward, C. Jason McKnight, Héloïse Lucaccioni, Iris Finci, Esther Kissling, Lia Sanodze.

**Visualization:** Mark A. Katz, Madelyn Yiseth Rojas Castro, Giorgi Chakhunashvili.

**Writing – original draft:** Mark A. Katz.

**Writing – review & editing:** Mark A. Katz, Madelyn Yiseth Rojas Castro, Giorgi Chakhunashvili, Nazibrola Chitadze, Caleb L. Ward, C. Jason McKnight, Héloïse Lucaccioni, Iris Finci, Tamila Zardiashvili, Richard Pebody, Esther Kissling, Lia Sanodze.

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
