## [Editor Report · Decision Letter 0]

15 Nov 2023

PONE-D-23-33035Primary Series COVID-19 Vaccine Effectiveness among Health Care Workers in the country of Georgia, March–December 2021PLOS ONE

Dear Dr. Katz,

Thank you for submitting your manuscript to PLOS ONE. After careful consideration, we feel that it has merit but does not fully meet PLOS ONE’s publication criteria as it currently stands. Therefore, we invite you to submit a revised version of the manuscript that addresses the points raised during the review process.

The Authors are expected to address all the criticisms by all Reviewers. In particular, please clarify if there’s overlap in the first sample and key informant, reassess quantitative statements (Reviewer #1), and further explain and assess the impact of the recruitment procedure, and consider the use of some basic statistics to strengthen the conclusion (Reviewer #2). In additional to the above comments, please address,

Could the authors provide the number of potential respondents invited and the number who refused the interview? This will help understand indirectly how potential respondents perceived the interview as voluntary. The authors are encouraged to provide other relevant information.The authors considered that the answers may be shorter for those who may feel uncomfortable to be interviewed, what would be the expected bias arising from this situation and how it will affect the main findings?Please provide more description and assessment on the interview process and further assess the potential direction and extent of bias due to the recruitment. Please submit your revised manuscript by Dec 30 2023 11:59PM. If you will need more time than this to complete your revisions, please reply to this message or contact the journal office at plosone@plos.org. Please include the following items when submitting your revised manuscript:A rebuttal letter that responds to each point raised by the academic editor and reviewer(s). You should upload this letter as a separate file labeled 'Response to Reviewers'.A marked-up copy of your manuscript that highlights changes made to the original version. You should upload this as a separate file labeled 'Revised Manuscript with Track Changes'.An unmarked version of your revised paper without tracked changes. You should upload this as a separate file labeled 'Manuscript'.

We look forward to receiving your revised manuscript.

Kind regards,

Eric HY Lau, Ph.D.

Academic Editor

PLOS ONE

Journal Requirements:

7. We notice that your supplementary figures are uploaded with the file type 'Figure'. Please amend the file type to 'Supporting Information'. Please ensure that each Supporting Information file has a legend listed in the manuscript after the references list.

Additional Editor Comments:

The Authors are expected to address all the criticisms by all Reviewers. In particular, please clarify if there’s overlap in the first sample and key informant, reassess quantitative statements (Reviewer #1), and further explain and assess the impact of the recruitment procedure, and consider the use of some basic statistics to strengthen the conclusion (Reviewer #2). In additional to the above comments, please address,

1. Could the authors provide the number of potential respondents invited and the number who refused the interview? This will help understand indirectly how potential respondents perceived the interview as voluntary. The authors are encouraged to provide other relevant information.

2. The authors considered that the answers may be shorter for those who may feel uncomfortable to be interviewed, what would be the expected bias arising from this situation and how it will affect the main findings?

3. Please provide more description and assessment on the interview process and further assess the potential direction and extent of bias due to the recruitment.

---

## [Decision Letter · Decision Letter 1]

23 Apr 2024

PONE-D-23-33035R1Primary Series COVID-19 Vaccine Effectiveness among Health Care Workers in the country of Georgia, March–December 2021PLOS ONE

Dear Dr. Katz,

Thank you for submitting your manuscript to PLOS ONE. After careful consideration, we feel that it has merit but does not fully meet PLOS ONE’s publication criteria as it currently stands. Therefore, we invite you to submit a revised version of the manuscript that addresses the points raised during the review process.

The Authors are expected to address all the criticisms by the Reviewer. In particular, please state the study objective clearly and add sample size considerations (Reviewer #1). In additional to the above comments, please address,

Serology, please make a clear statement about the ability of and how the serological tests were used to differentiate between SARS-CoV-2 infection, previous COVID-19 vaccination with mRNA and inactivated vaccines.Please make an assessment on how the above will affect the main study results especially for inactivated vaccine BBIBP-CorV.Results, the abbreviations VE and aVE were used interchangeably. Please harmonize and confirm that adjusted estimates are always presented.Line 276, there are some typos in the 95% CIs, for example remove “– “ in “(95%CI: 90; –98)” (Line 276), also in lines 277, 283, 292, 328 and others Please submit your revised manuscript by Jun 07 2024 11:59PM. If you will need more time than this to complete your revisions, please reply to this message or contact the journal office at plosone@plos.org. Please include the following items when submitting your revised manuscript:A rebuttal letter that responds to each point raised by the academic editor and reviewer(s). You should upload this letter as a separate file labeled 'Response to Reviewers'.A marked-up copy of your manuscript that highlights changes made to the original version. You should upload this as a separate file labeled 'Revised Manuscript with Track Changes'.An unmarked version of your revised paper without tracked changes. You should upload this as a separate file labeled 'Manuscript'.

We look forward to receiving your revised manuscript.

Kind regards,

Eric HY Lau, Ph.D.

Academic Editor

PLOS ONE

Additional Editor Comments:

The Authors are expected to address all the criticisms by the Reviewer. In particular, please state the study objective clearly and add sample size considerations (Reviewer #1). In additional to the above comments, please address,

1. Serology, please make a clear statement about the ability of and how the serological tests were used to differentiate between SARS-CoV-2 infection, previous COVID-19 vaccination with mRNA and inactivated vaccines.

2. Please make an assessment on how the above will affect the main study results especially for inactivated vaccine BBIBP-CorV.

3. Results, the abbreviations VE and aVE were used interchangeably. Please harmonize and confirm that adjusted estimates are always presented.

4. Line 276, there are some typos in the 95% CIs, for example remove “– “ in “(95%CI: 90; –98)” (Line 276), also in lines 277, 283, 292, 328 and others

Reviewers' comments:

Reviewer's Responses to Questions

**Comments to the Author**

1. If the authors have adequately addressed your comments raised in a previous round of review and you feel that this manuscript is now acceptable for publication, you may indicate that here to bypass the “Comments to the Author” section, enter your conflict of interest statement in the “Confidential to Editor” section, and submit your "Accept" recommendation.

Reviewer #1: (No Response)

2. Is the manuscript technically sound, and do the data support the conclusions?

Reviewer #1: Partly

3. Has the statistical analysis been performed appropriately and rigorously? 

Reviewer #1: Yes

4. Have the authors made all data underlying the findings in their manuscript fully available?

Reviewer #1: Yes

5. Is the manuscript presented in an intelligible fashion and written in standard English?

Reviewer #1: Yes

6. Review Comments to the Author

Reviewer #1: Introduction

Given tha rationale of the study, the introduction section lacks of a brief paragraph describing the precious role of health care workers in tackling COVID-19 pandemic. Moreoever, a further paragraph pointing out the importance of planning, coordinating and carrying out vaccination campaigns, in high-, middle-, and lower-income countries, should be anticipated in this section too.

Doing so, would allow the possibility to briefly state the benefits on the healthcare system (e.g., reduction of the pressure on the health care facilities, reduciton of the infections and overall mortality).

See some examples of references:

[Pascucci D, Nurchis MC, Sapienza M, et al. Evaluation of the Effectiveness and Safety of the BNT162b2 COVID-19 Vaccine in the Vaccination Campaign among the Health Workers of Fondazione Policlinico Universitario Agostino Gemelli IRCCS. Int J Environ Res Public Health. 2021;18(21):11098. Published 2021 Oct 22. doi:10.3390/ijerph182111098]

[Gaio V, Santos AJ, Amaral P, et al. COVID-19 vaccine effectiveness among healthcare workers: a hospital-based cohort study. BMJ Open. 2023;13(5):e068996. Published 2023 May 2. doi:10.1136/bmjopen-2022-068996]

[Arriola CS, Soto G, Westercamp M, Bollinger S, Espinoza A, Grogl M, Llanos-Cuentas A, Matos E, Romero C, Silva M, Smith R, Olson N, Prouty M, Azziz-Baumgartner E, Lessa FC. Effectiveness of Whole-Virus COVID-19 Vaccine among Healthcare Personnel, Lima, Peru. Emerg Infect Dis. 2022 Dec;28(13):S238-S243. doi: 10.3201/eid2813.212477. PMID: 36502444; PMCID: PMC9745240]

Lastly, it is of paramount importance to clearly describe what is the primary aim of the study at the end of the introduction section.

Materials and methods

It should be useful to better explicit, alongside the inclusion criteria, the exclusion ones.

There is no mention about sample size estimation. How was the sample size computed? Is it a purposive sampling? Please, specify these issues.

Discussion

It would be useful to discuss few main implications associated with vaccinating HCWs in light of the results about VE. This could have a significant impact on health systems, thus promoting tailored and targeted health policies should be clearly pointed out.

7. PLOS authors have the option to publish the peer review history of their article (what does this mean?). If published, this will include your full peer review and any attached files.

Reviewer #1: No

---

## [Author Response · Author response to Decision Letter 1]

14 Jun 2024

Dear Dr. Lau,

Thank you and thanks to the reviewers for the opportunity to respond to the reviewer's comments and your comments and revise our manuscript.

We have included a detailed response to reviewers in the attached Response to Reviewers document.

Sincerely,

Mark Katz

---

## [Editor Report · Decision Letter 2]

12 Jul 2024

Primary Series COVID-19 Vaccine Effectiveness among Health Care Workers in the country of Georgia, March–December 2021

PONE-D-23-33035R2

Dear Dr. Katz,

We’re pleased to inform you that your manuscript has been judged scientifically suitable for publication and will be formally accepted for publication once it meets all outstanding technical requirements.

Kind regards,

Eric HY Lau, Ph.D.

Academic Editor

PLOS ONE

Additional Editor Comments (optional):

Thanks for addressing all the editor’s and reviewers' comments. Congratulations on the excellent work!
---

## [Editor Report · Acceptance letter]

25 Jul 2024

PONE-D-23-33035R2 

PLOS ONE

Dear Dr. Katz, 

I'm pleased to inform you that your manuscript has been deemed suitable for publication in PLOS ONE. Congratulations! Your manuscript is now being handed over to our production team.

Kind regards, 

on behalf of

Dr. Eric HY Lau 

Academic Editor

PLOS ONE